# Sedentariness and Physical Activity during School Recess Are Associated with VO_2Peak_

**DOI:** 10.3390/ijerph17134733

**Published:** 2020-07-01

**Authors:** Fernando Calahorro-Cañada, Gema Torres-Luque, Iván López-Fernández, Elvis A. Carnero

**Affiliations:** 1Faculty of Humanities and Education Sciences, University of Jaen, 23071 Jaen, Spain; fernandocalahorro@gmail.com; 2Research Group CEDA (HUM-1016), Campus de las lagunillas, 23071 Jaén, Spain; 3Faculty of Education Sciences, University of Málaga, 29010 Malaga, Spain; ivanl@uma.es; 4Bioenergetics and Exercise Laboratory, Translational Research Institute AdventHealth, 301 E. Princeton St, Orlando, FL 32804, USA; Elvis.AlvarezCarnero@Adventhealth.com

**Keywords:** adolescent, physical activity, school

## Abstract

Recess time (RT) is a main component of school-based activities, and could contribute up to 40% of the physical activity (PA) recommended in the health guidelines. The main goal was to analyze the association between accelerometer-measured PA and sedentary time during RT with cardiorespiratory fitness (CRF). A total of 146 children and adolescents, aged 8–19 years, were recruited from six schools. PA levels were measured with GT3X accelerometers over 7 days. CRF was measured using a portable breath-by-breath gas analyzer. A general linear model (GML) was conducted to analyze the association between PA intensities and CRF during RT. Additionally, a logistic binary regression was used to evaluate the risk of unhealthy CRF among different categories of PA and sedentary time. Participants classified as healthy showed higher PA levels during RT than those classed as unhealthy. GML analysis showed that sedentary time during RT was significantly associated with VO_2Peak_. Finally, compared to individuals accumulate less than 15-min of ST during recess, individuals who were sedentary for more than 15 min during RT presented 43.78 times of having unhealthy CRF (95% CI 3.873–494.824). Our data suggest an association between recess sedentary time and unhealthy CRF. Thus, school-aged children and adolescents must be empowered to perform PA during RT to prevent the deleterious effects of sedentary time on CRF.

## 1. Introduction 

Cardiorespiratory fitness (CRF) measured as peak oxygen consumption (VO_2peak_) has been described as a strong marker of health among children and adolescents [1] and independent predictor of disease and mortality in adult and older populations [2]. It is also well-known that regular moderate to vigorous physical activity (MVPA) has a positive influence on cardiometabolic risk factors [3]. Additionally, sedentary time (ST) may be associated with CRF (measured as maximum oxygen consumption) independently of total physical activity (PA) [4], which means MVPA and ST have independent effects on health outcomes [5]. The role of sedentary behavior has been recently described in the literature, such that excessive sedentary behavior has been linked with cardiovascular disease and low CRF (measured as maximum oxygen consumption) independently of MVPA [4,6]. Breaking up ST with small bouts of activity has been associated with improved metabolic profile in adults [7]. These deleterious effects and the high prevalence of ST have also been described in children and adolescents [8,9].

PA intervention in children and adolescents has been shown to be effective in decreasing ST. These intervention programs consisted of PA bursts or reduced length-bouts [10,11,12], physical education [13] or exercise training during school-based interventions. Others [14] indicated that school interventions are at least as effective as out-of-school interventions in increasing PA and decreasing ST. These previous findings provided empirical and scientific knowledge to support school scheduling, with regards to setting aside valuable time for promoting actual health outcomes. In this regard, recess time (RT) is a daily opportunity where students can freely choose their activities, and it has been reported to contribute up to 40% of recommended daily PA [15,16]. Several intervention programs during recess have been shown to increase PA [17] and cognitive performance [18]. Additionally, recent studies have demonstrated that a multi-component PA intervention during recess can decrease ST and increase light PA by 5.9% [19]. Despite this positive contribution, the influence of PA and ST, during an unstructured activity such as RT, on health outcomes as VO_2Peak_ remains not completely clear in students of Middle School age. Moreover, the ST of RT has been rarely reported on [15,19,20,21], and much less has its deleterious association with VO_2Peak_ [22] been analyzed, none of which was found with Middle School students. For instance, cutoffs in healthy and unhealthy CRF profiles for children and adolescents have been associated with cardiovascular or metabolic markers [1,4,23], but there is a lack of knowledge regarding how PA or ST can influence the risk of having heathy (H) or unhealthy (UH) VO_2Peak_.

Since RT represents a significant weekly amount of time available to expend in free PA, the influence of MVPA and ST during RT on VO_2Peak_ is plausible. Nevertheless, studies have been largely focused on general weekly PA, not measuring specific PA contexts such as RT [6,24,25,26]. So, whether PA or ST during recess can make a significant contribution to attaining a higher VO_2Peak_, or to having a H/UH VO_2Peak_, is a question that has not been answered in the literature yet. 

The purpose of this study was to examine whether objectively measured recess PA and ST are associated withVO_2Peak_ independently of total daily PA. Additionally, we hypothesize that PA or ST variables could partially determine UH or H VO_2Peak_ condition in children and adolescents.

## 2. Materials and Methods 

### 2.1. Participants

In this cross-sectional study, 146 healthy Spanish 3rd–12th grade children and adolescents, aged 8–19 years, were recruited from 6 schools and High Schools in Spain (Malaga and Jaen regions). The years of data collection were 2011 to 2013. Detailed information about procedures and risks was given to parents/guardians and a written informed consent was obtained prior to participation in the study. All procedures were in accordance with the ethical principles for medical research involving human subjects from the Helsinki Declaration [27], and approved by the Ethical Committee of the Sports Medicine School (University of Malaga, reference: EMEFIFE-3-2011), Spain. This protocol was additionally reviewed and followed the ethical requirements of grant application opportunities laid out by the Spanish Ministry of Economy and Competitiveness (Plan Nacional I+D+I 2008-2011).

### 2.2. Procedures

A total of 668 accelerometers (ACLs) were distributed and worn, but only 282 participants had valid ACL data, and 386 had either invalid ACL data (lost data 33 ACLs), lost device (3 ACLs) or did not meet the Troiano [28] criteria of ≥10 valid wear hours per day (350 ACLs), which represents an attrition rate of 57.8%. After considering the classical inclusion criteria, we filtered the data for selecting only those with at least 4 valid recess days (172 valid ACL recordings). Regarding CPX-T, 503 breath-by-breath tests were performed until volitional exhaustion; although only 480 were considered valid (85 were discarded because of telemetry errors, non-valid data or technical issues). All together, 172 participants had accelerometry and CPX-T valid data; however, 26 were removed from the dataset due to geographical bias (the weather in the North region could limit the amount of PA during recess). We finally included 146 scholars (Figure 1), which represents 21.85% of the total accelerometers; although this was not due to low compliance, but related to strict inclusion criteria to obtain valid data. Peak oxygen consumption (VO_2Peak_) and age were similar between non-selected and selected participants (40.45 vs. 39.3 mL/kg/min, t = −1.29, *p* = 0.199; 13.3 vs. 13.1 years, t = 0.536, *p* = 0.592 non-selected/selected, respectively); also, the proprotions of female/male (48.1% vs. 42.7% females, non-selected/selected, respectively; Chi-square = 1.236, *p* = 0.266) and students without and with overweight/obesity (30.0% vs. 30.7% students with overweigth/obesity, non-selected/selected, respectively; Chi-square = 0.020, *p* = 0.887) were similar. A binary variable was utilized to classify the participants according to BMI categories (normal weight and overweight/obese) [29].

Actigraph GT3X ACLs were used to measure PA (Actigraph, Pensacola, FL, USA), which collects motion data on 3 orthogonal axes, which ensured reliability in quantifying PA [30]. During the first study day, for each participant, a research team member set up the ACL and provided written ACL information. Students were instructed to wear the ACLs for 7 consecutive days, except during water activities (swimming, showering) or while sleeping [31,32]. ACLs were programmed to record the entire day (0.00 a.m. to 23.59 p.m.). A weekly activity diary was given to record when they went to and got out of bed, RT and water activities. Standard duration of recess was 30 min for every school day. A wear-time of ≥10 h/day for ≥5 weekdays (4 weekdays, 1 weekend day) was used as the criterion for valid measurement [28], therefore only participants who had at least 4 of the 5 weekly recesses were included in the analysis. The data collection protocol has been described elsewhere [33]; briefly, participants wore ACLs next to the right hip [34], fastened with an elastic belt. Data was collected for 7 consecutive days with 1 s epoch, as in previous studies with children between 5 and 16 years [30,35], to accurately capture patterns of high intensity and short-duration PA. Data was downloaded and analyzed using the software ActiLife 6.0 (ActiGraph). The cut-off points proposed by Evenson, Catellier, Gill, Ondrak and McMurray [34] were used: Sedentary PA ≤ 100; Light PA ≥ 100; Moderate PA ≥ 2296 and Vigorous PA ≥ 4012 counts·min^−1^. These limits were recently validated for adolescents [35]. All PA levels were reported as minutes/day or daily percentages in order to standardize PA values. Total PA (TDPA) was calculated by adding light plus MVPA. 

VO_2Peak_ was assessed using a portable breath-by-breath metabolic unit (Metamax 3B, Cortex Biophysic, Leipzig, Germany), which has been previously validated [36], while performing the Chester Step Test on a bench [37]. The gas analyzers were calibrated using room air and a gas tank with known concentrations of O_2_ and CO_2_ (5.0% CO_2_, 15.0% O_2_). The volume sensor was calibrated with a 3-litre volume syringe (Calibration syringe D, Sensormedics Hans Rudolph Inc 5530) each testing day. Heart rate was simultaneously recorded using a Polar Team (Finland). During the test, children were encouraged to exercise until exhaustion, and the test was terminated either by the subject (because of dyspnea and/or leg fatigue) or by the supervisor when the student was unable to maintain proper cadence for 15 s [38]. Data from indirect calorimetry was collected using Metasoft v. 1.11.05 software (Cortex Biophysic, Leipzig, Germany) and averaged in 30-s periods. All tests were supervised by a sports medicine physician. 

### 2.3. Statistical Analyses

All variables are shown as mean and SD or median and ranges. The Kolmogorov–Smirnov test was used to confirm a non-parametric distribution. Spearman’s correlation analyses were carried out to explore associations between VO_2Peak_ and PA intensities during RT. In order to identify which recess PA variables were significantly associated with the outcome variable (VO_2Peak_ as dependent variable), a general linear model adjusted to daily ST and MVPA time without RT was utilized [sex, age or school group (Primary/Middle/High School)], and BMI and wear time were also included as co-variables.

Additionally, children and adolescents were classified as “High VO_2Peak_/Healthy” (H) or “Low VO_2Peak_/Unhealthy” (UH) according to sex-specific and age-specific cut-off criteria for participants between 10 and 18 years [23] and under 10 years [1]. Mann–Whitney tests were conducted to compare PA variables between sexes and VO_2Peak_ categories. The sample was also classified as Elementary School (<10.49 years old), Middle School (10.50 to 14.49 years old) and High School (≥14.50 years old). A Kruskall–Wallis test for comparing medians among age category was conducted. Finally, a binary logistic regression (adjusted for age, sex and daily TDPA without RT) was conducted between recess PA variables and VO_2Peak_ categories. For this purpose, recess PA variables were transformed into binary variables as follows: (i) scholars who met or did not meet 15 min of ST during RT; (ii) scholars who met or did not meet 5 min of MVPA (5 min MVPA). The level of significance was set at *p* ≤ 0.05 for all the tests. Analyses were conducted with the SPSS software (v. 20.0).

## 3. Results

Participants included in the analysis were 13.16 years old on average, and 69.2% were normal weight, without difference between girls and boys (Table 1). Neither girls nor boys met PA activity recommendations on average, although boys performed significantly more minutes of MVPA than girls (45.7 vs. 52.2, *p* ≤ 0.05) and had a higher VO_2Peak_ than girls. Additionally, group ages revealed differences regarding age among all groups (*p* ≤ 0.001). Regarding VO_2Peak_, group ages showed differences between Elementary and Middle School (41.93 vs. 44.23 mL/kg/min, *p* ≤ 0.05), and between Middle School and High School (44.23 vs. 33.99 mL/kg/min, *p* ≤ 0.01). Finally, total ST was lower in Elementary compared to Middle School (843.34 vs. 925.39 min/day, *p* ≤ 0.05), and even lower in High School and Middle School (925.39 vs. 624.17 min/day, *p* ≤ 0.01). 

Overall, H participants and boys were more active during RT, and less sedentary than the UH participants and their female counterparts (both min/day and %). Regarding VO_2Peak_ categories, children and adolescents in the H category had significantly higher PA levels than UH (*p* ≤ 0.001, Table 2), and lower BMI (19.5 vs. 20.8; *p* ≤ 0.05). Additionally, no differences were revealed among sexes within the H category. Regarding the UH category, boys from the UH group accrued a higher percentage of time during RT to contribute to total MVPA than UH girls (6.27% vs. 4.19%; *p* ≤ 0.005), and MVPA time during recess (3.76 min/recess vs. 2.51; *p* ≤ 0.005).

We found signigicant correlations adjusted for TDPA between VO_2Peak_, BMI and PA intensities during RT; all PA intensities during RT were positively associated with VO_2peak_ (r = 0.417, r = 0.352, r = 451, r = 0.464 for light, moderate, vigorous and total recess PA, respectively; *p* < 0.001 for all). Negative associations were found between VO_2Peak_ and BMI, and ST (−0.180, *p* ≤ 0.05; −0.471, *p* ≤ 0.001, respectively). The general linear model adjusted to daily ST and MVPA time without RT was utilized [sex, age or school group (Primary/Middle/High School)], and BMI and wear time were also included as co-variables).

Our general linear model (Table 3) showed that ST during RT, BMI and age were inversely associated with VO_2Peak_. Sex was also a significant predictor of VO_2Peak_. The final model, adjusted for daily ST and daily MVPA time without RT, explained 40.4% of the variance of VO_2Peak_ (adjusted R^2^ = 0.404, *p* < 0.001). None of the indivual PA intensities during RT are entered in the final model. A deeper analysis of our data by school level (Primary/Secondary/High School) reveled a significant interaction between recess ST, school level and sex; the relationship between sedentary time and VO_2peak_ had a significantly lower beta coefficient in girls than boys attending High School (beta = −0.78; t = −3.709, *p* < 0.001). No significant interactions were found in the other school level groups. The model including school level groups had a lower predicted power than the one including age (adjusted R^2^ = 0.287 vs. adjusted R^2^ = 0.404). 

The binary logistic regression (adjusted for age, sex and daily TDPA without recess time) was conducted between recess ST and PA variables, and VO_2Peak_ categories. For this purpose, recess PA and ST variables were transformed into binary variables, as follow: (i) scholars who achieved or did not achieve 15 min of ST during RT; and (ii) scholars who achieved or did not achieve 5 min of MVPA (5 min MVPA). Odds ratios (OR) adjusted from the logistic regression between VO_2Peak_ categories and categorical variables of recess PA/ST confirmed that, compared to individuals who accumulated less than 15 min of ST during recess, individuals who accumulated more than 15 min of ST during recess had 43.78 times higher risk of unhealthy cardiorespiratory fitness (Table 4). Moreover, the MVPA category (>5 min) was not a significant predictor of H CRF (OR = 1.66e^9^; Wald = 0, *p* = 0.998). 

## 4. Discussion

In this paper, we analyzed PA and ST levels during RT, wherein our primary aim was to examine associations between PA/ST and VO_2Peak_ during RT. The main finding was that low levels of ST during recess were associated with higher values of CRF for children and adolescents. We found a higher likelihood of having an UH VO_2Peak_ when spending more than 15 min on sedentary behavior during RT. To our knowledge, there is no previous evidence reporting these associations between recess ST (measured with ACLs) and VO_2Peak_. Nonetheless, the effects of ST on impaired VO_2Peak_ have been reasonably well established. For example, prolonged sitting time during childhood has been inversely related with CRF (measured as maximum oxygen consumption and Vo2_Peak_) [39,40]. Additionally, interruptions of ST with short breaks of PA is a useful strategy for promoting health outcomes [6,7,41,42]. We did not explore these kinds of responses; therefore, we could not confirm whether specific sedentary/PA patterns or profiles influenced our results. Our hypothesis is that spending more than 50% of the recess (15 min) on sedentary behavior will increase the probability of having UH VO_2Peak_, whether this ST is accrued in a 15-min bout or 1-min bouts. Previous studies have described both short or long bouts as providing similar benefits for VO_2Peak_ [11], which may support our hypothesis. Altogether, these results confirmed that encouraging children to interrupt ST (with bursts of PA) is a viable strategy to ensure a healthy VO_2Peak_ in scholar settings. Nevertheless, we could not explore the underlying mechanisms that explain why our participants broke their sedentary behavior during RT, which is as yet poorly understood.

In addition to the relevance of ST, our general linear model analysis confirmed the importance of BMI and age as significant predictors of VO_2Peak_ during childhood and adolescence. Intuitively, age should be positively associated with VO_2Peak_, because it is related with determinants of absolute maximum oxygen uptake, like fat free mass or hemoglobin [43]. However, it was paradoxical that age was negatively associated with VO_2Peak_. We reported VO_2Peak_ relative to weight (ml/kg /min), and this could confound the interpretation of the results due to growth-associated changes in weight with age [44]. The mismatch of weight increase and cardiovascular function may be responsible for a steady or an even reduction in VO_2peak_ per kg throughout childhood and adolescence [43,45]. Nevertheless, during growth, the rate of decline of relative VO_2Peak_ may also be affected by inactivity. Our partial analysis of school groups showed that girls at High School age exhibited a larger reduction in VO_2peak_ per minute of recess ST than boys. This latter result could be indicative of a sexual dysmorphism with regard to the effect of ST on cardiorespiratory fitness after puberty [45]; nevertheless, this is speculative, and our results must be confirmed in specific maturational studies. 

As previously reported by others [46], we found a relationship between BMI and cardiorespiratory fitness after adjusting to TDPA [46]. Consequently, the deleterious effect of BMI on cardiovascular fitness may be independent of the volume of PA (e.g., genetics or nutrition). These results support the high significance of keeping a low BMI to preserve metabolic health [1]. In a previous study, Veijalainen et al. [22] investigated associations of CRF, PA, sedentary behavior and body fat percentage with arterial stiffness and dilation capacity in prepubertal children. They concluded that a combination of a low CRF and a low unstructured PA and high body fat have the strongest associations with detrimental health outcomes (arterial stiffness). These results support the relevance of our data, as recess time involves typically unstructured PA and could be an indicator of daily unstructurated PA. 

The novel finding of this study was that ST during RT was negatively associated with VO_2Peak_ independently of TDPA; however, it was surprising that MVPA during RT was not related with VO_2Peak_. This result may suggest that the minimum threshold of PA necessary to keep a healthy VO_2Peak_ must be different from the one necessary to have a high VO_2Peak_. Specific physiological mechanisms related with sedentariness have shown that total daily ST may be an independent contributor to cardiovascular risk [7].

The strengths of this research were that VO_2Peak_ was directly measured by indirect calorimetry, and PA and ST were assessed using GT3X accelerometers (at 1 s epochs) in two Spanish regions (including six Middle and High Schools). Furthermore, at least four of the five recesses were included as inclusion criteria, which increased the reliability of observed PA/ST behaviors during RT and the power of the statistical tests. Children and teachers were informed about the goal of wearing the ACL, but without mentioning the importance of being active during RT or any other time of the day, which might have avoided a biased behavior.

There are some limitations that may make us interpret our results cautiously. The sample size was a limitation for some of the statistical analysis of this study. Sample size was drastically reduced because ~80% did not meet the tight inclusion criteria. Regarding the PA measurement, accelerometers have been shown to be valid and reliable, but they have some limitations. These devices do not recognize the type of PA behavior (sitting, standing still, strength activities, etc.), and partial quantification mistakes of sedentary/activity times could have been possible due to an elevated number of manual activities or isometric activities. Despite these limitations, we do not believe these activities constitute a significant amount of the recess PA. As the study sample consists of a convenience sample of six schools in two regions of Spain, the generalizability of the results to the Spanish population cannot be inferred. The PA data during RT (Table 2) were similar to the MVPA reported for English children [47]; however, the percentage of MVPA was lower for children and adolescents from England than for those from Spain [16,48], which suggests our participants were poorly active, as confirmed by the low accomplishment of the MVPA guidelines that recommend 60 min a day (Table 1). These differences might be consequences of several reasons. On one hand, PA behavior could vary depending on social characteristics and the boys/girls ratio [49]. On the other hand, those differences may involve different sport playgrounds, space availabilities and access to sport equipment, which have also been suggested to influence PA behavior [49]. That said, it does not seem probable that social or environmental characteristics affect our results, since this study was conducted only in public Primary and Secondary Schools allocated in middle class neighborhoods in different regions of southern Spain. In a post hoc analysis, we discovered the unsually low participation (prevalence) in recesses activity of participants from northern Spain, which could be related to a higher number of rainy days in this region during the assessment week (see methods section). We also cannot confirm if the results from other regions of Spain or Europe were obtained in similar circumstances and settings. Regarding sex, we had a well-balanced boys/girls ratio, and in accordance with previous studies, boys were more active than girls during RT [16,48,49,50]. Finally, the cross-sectional nature of this study should be considered, from which we cannot infer causality between variables. Future studies should extend the assessment of patterns of children’s PA/ST into recess, its associations with VO_2Peak_, different school contexts, longitudinal studies and PA interventions.

## 5. Conclusions

In summary, the main finding of this study suggested that children or adolescents devoting more than 15 min of RT to ST had a higher likelihood of being UH than those who were less sedentary during RT. Additionally, our results extended the evidence of the influence of adiposity (BMI) and age on VO_2Peak_ in children and adolecents.

## Figures and Tables

**Figure 1 ijerph-17-04733-f001:**
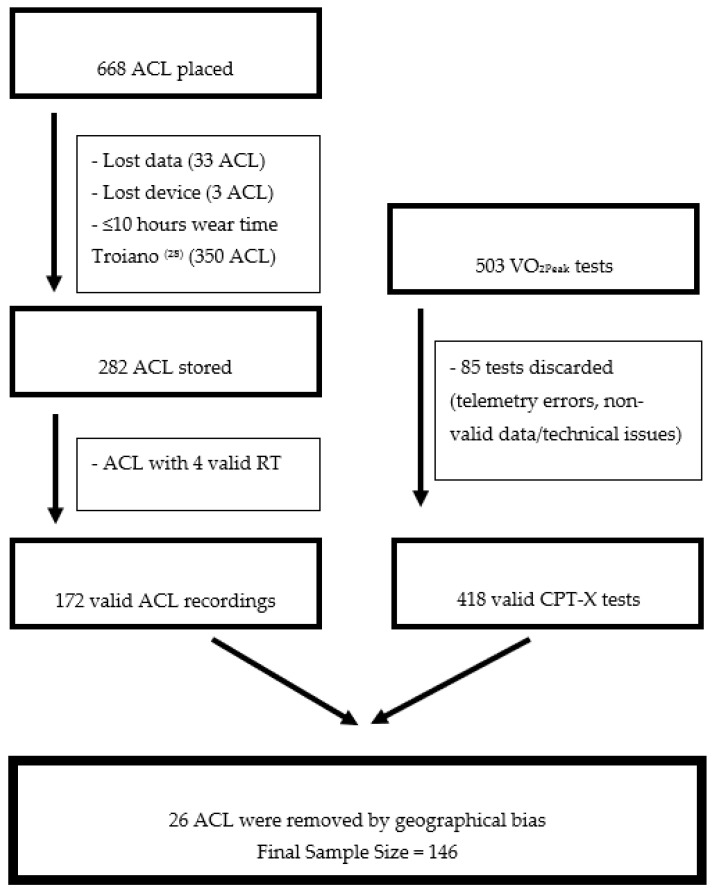
Flux diagram of inclusion criteria of the sample.

**Table 1 ijerph-17-04733-t001:** Descriptive characteristics of the sample.

Variable		Total (n = 146)			Girls (n = 64)			Boys (n = 82)			Sig.	Elementary School (n = 17)			Middle School (n = 79)			High School (n = 50)			Sig.	1 vs. 2	1 vs. 3	2 vs. 3
		Mean	SD		Mean	SD		Mean	SD			Mean	SD		Mean	SD		Mean		SD				
Age	(years)	13.16	±	2.42	13.26	±	2.40	13.09	±	2.46		9.67	±	0.69	12.17	±	1.27	15.93	±	0.99	***	◊◊◊	^^^	+++
NW	(%)	69.2			73.4			65.9				52.9			64.4			82.0						
BMI	(kg/m^2^)	20.21	±	3.74	19.83	±	3.81	20.5	±	3.68		19.59	±	3.34	20.13	±	3.95	20.55	±	3.57				
VO_2Peak_	(ml/kg/min)	40.45	±	10.31	35.85	±	8.6	44.05	±	10.16	***	41.93	±	10.31	44.23	±	9.34	33.99	±	8.65	***	◊		+++
Sedentary	(min/day)	813.98	±	289.90	788.72	±	273.42	833.38	±	302.17		843.34	±	251.86	925.39	±	317.36	624.17	±	99.00	***	◊		+++
MVPA	(min/day)	49.43	±	19.36	45.74	±	17.49	52.26	±	20.33	*	47.13	±	13.27	48.69	±	18.36	51.42	±	22.61				
TDPA	(min/day)	150.61	±	52.98	143.63	±	52.21	155.97	±	53.25		134.98	±	36.47	143.26	±	41.23	164.75	±	70.04				
PA Rec	(%)	36.3			31.3			40.2				41.2			32.9			40.0						

BMI: Body Mass Index; NW: Percent of children and adolescents with normal weight; VO_2Peak_: Oxygen peak uptake; MV: Moderate to vigorous; TDPA: Total daily physical activity; PA Rec: Meeting PA health 60 min guidelines. Girls vs. Boys Mann–Whitney U test for comparing medians among sexes, *: *p* ≤ 0.05; ***: *p* ≤ 0.001. Kruskall–Wallis for comparing medians among age category, ***: *p* ≤ 0.001; Mann–Whitney U test for comparing medians among age groups: 1 vs. 2 (Elementary vs Middle School); ◊: *p* ≤ 0,05; ◊◊◊: *p* ≤ 0,001; 1 vs. 3 (Elementary School vs. High School) ^^^: *p* ≤ 0.001; 2 vs. 3 (Middle School vs. High School), +++: *p* ≤ 0.001.

**Table 2 ijerph-17-04733-t002:** Daily physical activity levels during recess between boys and girls split by unhealthy and healthy VO_2Peak_.

Variable		Total			Unhealthy			Healthy			Sig.	Girls			Boys			Sig.
		(n = 146)			(n = 75)			(n = 71)				(n = 64)			(n = 82)			
		Mean		SD	Mean		SD	Mean		SD		Mean		SD	Mean		SD	
VO_2Peak_	(ml/kg/min)	40.45	±	10.31	32.17	±	4.93	49.2	±	6.6	***	35.85	±	8.6	44.05	±	10.16	+++
Sedentary	(min/recess)	21.45	±	5.75	24.38	±	3.72	18.36	±	5.92	***	23.08	±	4.29	20.18	±	6.42	++
Light	(min/recess)	6.32	±	3.07	4.95	±	2.29	7.75	±	3.14	***	5.61	±	2.6	6.87	±	3.3	+
Moderat	(min/recess)	1.93	±	1.58	1.32	±	0.98	2.57	±	1.84	***	1.58	±	1.07	2.2	±	1.85	
Vigorous	(min/recess)	1.39	±	1.63	0.63	±	0.75	2.2	±	1.9	***	1.02	±	1.06	1.68	±	1.92	
MVPA time	(min/recess)	3.32	±	3.03	1.95	±	1.59	4.77	±	3.49	***	2.6	±	2.01	3.88	±	3.54	
Total P	(min/recess)	9.64	±	5.61	6.91	±	3.35	12.52	±	6.08	***	8.21	±	4.14	10.75	±	6.34	+
Sedentary	(%)	69.09	±	17.87	78.06	±	10.17	59.61	±	19.36	***	73.94	±	12.7	65.3	±	20.33	++
Light	(%)	20.24	±	9.66	15.67	±	6.71	25.06	±	10	***	17.75	±	7.76	22.18	±	10.56	++
Moderate	(%)	6.19	±	5.07	4.21	±	3.06	8.27	±	5.89	***	5.02	±	3.33	7.1	±	5.96	
Vigorous	(%)	4.49	±	5.23	2.05	±	2.48	7.06	±	6.1	***	3.3	±	3.42	5.42	±	6.16	
MVPA	(%)	10.67	±	9.71	6.26	±	5.08	15.33	±	11.18	***	8.31	±	6.38	12.51	±	11.37	
Contr. Guid	(%)	5.54	±	5.05	3.25	±	2.65	7.95	±	5.82	***	4.34	±	3.35	6.47	±	5.9	
Contr. MVPA	(%)	6.97	±	5.91	4.07	±	2.88	9.99	±	6.71	***	6.11	±	4.55	7.63	±	6.72	
Steps	(steps/recess)	600.48	±	449.49	381.56	±	250.8	831.73	±	496.41	***	496.54	±	325.96	681.6	±	513.66	

BMI: Body Mass Index; VO_2Peak_: Oxygen peak uptake; PA: Physical activity; MV: Moderate to vigorous; Total PA: Physical activity (Light + Moderate + Vigorous); Contr. Guid: Contribution to 60 min health guidelines; Contr. MVPA: Contribution to total MVPA; Guid: Guidelines. Girls vs. Boys, Mann–Whitney U test for comparing medians among sexes, +: *p* ≤ 0.05; ++: *p* ≤ 0.01; +++: *p* ≤ 0.001. Healthy vs. unhealthy students, Mann–Whitney U test for comparing medians among VO_2Peak_ Status, ***: *p* ≤ 0.001.

**Table 3 ijerph-17-04733-t003:** Univariate general linear model to estimate VO_2peak_ predictors.

Variables	Unit	Beta	SE		Sig.	(95% CI)
Intercept		70.4	±	8.021	<0.001	(54.53–86.25)
Age	(years)	−0.966	±	0.313	0.002	(−1.59–−0.35)
ST_woRecess	(min/day)	0.005	±	0.003	0.087	(0.00–0.01)
MVPA_wo Recess	(min/day)	−0.002	±	0.036	0.966	(−0.07–0.07)
ST Recess	(min/recess)	−0.485	±	0.132	<0.001	(−0.75–−0.22)
Sex (female)		−6.67	±	1.384	<0.001	(−9.40–−3.93)
BMI	(kg/m^2^)	−0.38	±	0.178	0.034	(−0.74–−0.03)

SE: standard error; ST_woRecess: average total daily sedentary time not considering ST recess; MVPA_woRecess: moderate vigorous physical activity not including MVPA during recess; ST Recess: sedentary time during reces; BMI: Body Mass Index.

**Table 4 ijerph-17-04733-t004:** Binary logistic regression for healthy/unhealthy VO_2Peak_.

Variables	Unit	Wald	Sig.	OR	(95% C.I. for OR)
ST_woRecess	(min/day)	15.189	<0.001	1.004	(1.007–0.996)
Age	(years)	11.921	0.001	0.660	(0.835–1.516)
ST Recess > 15 min		9.329	0.002	43.78	(3.873–494.824)
BMI category (OW/OB)		9.114	0.003	0.139	(0.039–0.501)
Sex (male)		7.488	0.006	4.026	(1.485–10.917)
Constant		1.553	0.213	0.061	

Reference category was unhealthy VO_2Peak_. OR: Odds ratio; CI: confidence intervals; ST_woRecess: average total daily sedentary time not considering ST recess; ST Recess: Sedentary time during recess; BMI category: binary body mass index category (overweight or obesity is the reference). Variables that did not enter in the model: Total daily physical activity without PA recess time.

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
