# Peer review of "Sedentariness and Physical Activity during School Recess Are Associated with VO2Peak"

_ijerph, 2020, doi:10.3390/ijerph17134733_

Round 1

Reviewer 1 Report

The recess time is very important to promote the daily activity, which may make a significant contribution for improving the health-related outcomes. And, cardiorespiratory fitness is a strong marker for the whole health. Therefore, the topic of this study is of great practical significance. But, for the manuscript, I have made some comments and recommendations as below where the authors may want to consider.

1. Title

Physical activity and sedentary are the two different terms. Actually,  this study focused on "could recess activities and sedentary time be relevant to cardiovascular health". Therefore, the title needs to be modified.

2. Abstract

  1. The aim of this study should be “To analyze the association between accelerometer-measured physical activity and sedentary time during recess time with cardiorespiratory fitness”, not just the association between physical activity and cardiorespiratory fitness.
  2. The value of the odd ratio (OR) should be greater than 0, furthermore, your confidence interval did not include the value of the point estimate. Please double check your data.
  3. The word of "effect" (line 26) is not appropriate in this cross-sectional study.

3. Introduction

  1. The rationale in paragraph 2 is weak (Lines 42-57).
  2. Do you mean PA interventions could increase sedentary time (Line 44)?
  3. What is the definition of older students in line 52, is that means the association between activity in RT and cardiorespiratory fitness has been conducted in young student, but there is no citation in your manuscript.
  4. The “association” is more appropriate than the “influence” in this cross-sectional study in line 56.

4. Methods

  1. As the important independent variable in this study, the number of recess activities per day and the amount of per recess should be detailed in the method.
  2. Please present the approval number of ethics committee, if you have.
  3. Please use the full name for ACL for the first time in line 76.
  4. The cut points for different physical activity intensities you selected were made for children aged 5-8 years old (Ref. 31). Is it suitable for your participants aged 8-19 years old?
  5. How do you classify the normal weight and non-normal weight should be detailed in the method.
  6. In your study, should the VO2max be the dependent variable (line 108)?

5. Results

  1. My opinion is that if the moderate to vigorous physical activity (MVPA) has been reported, it is not necessary to report MPA and VPA separately again.
  2. Total MVPA time in Table 2 should be 3.4 min/recess rather than 2.7 min/recess, please double check all of your data. Moreover, the term should be consistent in the whole manuscript, for example, MVPA was used in Table 1 and 3, but MV in table 2.
  3. I think the title “The association between BMI, sedentary time and physical activity with VO2max” is clearer for Table 3. In addition, should the unit of most of the indicators in table 3 be min/recess rather than min/day.
  4. I recommend you present the results of general linear model, which are more important than the results of correlation in Table 3. Furthermore, in order to obtain the independent association, other independent variables need to be adjusted. For example, daily ST and daily MVPA time without RT, and physical activity during RT (including LPA and MVPA) need to be adjusted, if you want to know the independent association between ST during RT and cardiorespiratory fitness (The publication PMID: 29221862 can be your reference).
  5. Considering age and sex are two significant predictors for VO2max, why performed the data analysis with different sex and ages together for table 3 and 4.
  6. For the table 4. a) The reference category for each independent variable need to be presented; b) the outcome indicator for the table 4 need to be noted (“High VO2max/ Healthy” or “Low VO2max / Unhealthy”); c) OR value cannot be negative, please double check your data; d) Health VO2max should be your outcome rather than the reference category.

6. Discussion

The discussion section is weak. Furthermore, because of the main results probably are incorrect and need to be updated, the discussion may need to be rewritten.

Author Response

Dear editor, thank you very much for allowing to re-submit a revised version of the article. We have tried our best to accommodate and address the reviewers’ comments (please see text in changes mode).

REV.- The recess time is very important to promote the daily activity, which may make a significant contribution for improving the health-related outcomes. And, cardiorespiratory fitness is a strong marker for the whole health. Therefore, the topic of this study is of great practical significance. But, for the manuscript, I have made some comments and recommendations as below where the authors may want to consider.

AUT.- We really appreciate all your inputs and time to review our manuscript, which certainly will improve the quality of this work.

REV.-  Title

Physical activity and sedentary are the two different terms. Actually, this study focused on "could recess activities and sedentary time be relevant to cardiovascular health". Therefore, the title needs to be modified.

AUT.- Thanks for the suggestion. An alternative title has been suggested.

REV.-  Abstract

The aim of this study should be “To analyze the association between accelerometer-measured physical activity and sedentary time during recess time with cardiorespiratory fitness (CRF)”, not just the association between physical activity and cardiorespiratory fitness

AUT.- We modified the aim following your recommendation.

REV.-  The value of the odd ratio (OR) should be greater than 0, furthermore, your confidence interval did not include the value of the point estimate. Please double check your data.

AUT.- Your appreciation is partially correct. As you well said simple OR are always values above zero; however, our OR come from a logistic regression (logit OR), so they could be lower than 1 (Kleinbaum DG, Kupper LL, Muller KE, Nizam A. Dummy Variables in Regression. Edtion ed. In: Kugushev A, ed. Applied Regression Analysis and Other Multivariable Methods: Duxbury Press, 1998:317-60). In this manuscript, we preferred to use logit ORs to provide information about the direction of the relationship between dependent and independent variables. To avoid confusion and provide a better understanding, we included the full binary logistic regression results with logitORs and ORs for all variables.

We recognize we should have provide a better description of the statistic in the table.

REV.-  The word of "effect" (line 26) is not appropriate in this cross-sectional study.

AUT.- We changed “effect” by “association”.

REV.-   Introduction

The rationale in paragraph 2 is weak (Lines 42-57).

AUT.- This has been rewritten.  

REV.-  Do you mean PA interventions could increase sedentary time (Line 44)?

AUT.- Thanks for noticing. We changed appropriately (“decrease”). 

REV.-  What is the definition of older students in line 52, is that means the association between activity in RT and cardiorespiratory fitness has been conducted in young student, but there is no citation in your manuscript.

AUT.- This refers to Van Kann’s study [1](reference 19 in manuscript), which was conducted in children ranged from 8 to 11 years. The second paragraph of the introduction has been modified to fix all mistakes (please, see modifications in the manuscript).

REV.-  The “association” is more appropriate than the “influence” in this cross-sectional study in line 56.

AUT.- Thanks, it has been modified.

REV.-  Methods

As the important independent variable in this study, the number of recess activities per day and the amount of per recess should be detailed in the method.

AUT.- One recess per day is carried out in all the places of the study, which consisted of 30 minutes. The activities during the recess were free and each student selected/made her/his own choices. Nevertheless, an additional analysis by geographical regions made us to remove the students from the North region due to a significant difference in the weather between North and South regions (more rainy days in the North than in the South); this aspect could bias the amount of PA  and number of students who went outdoors during the recess time (the results did not change significantly).  Additional information has been included in the manuscript.

REV.-  Please present the approval number of ethics committee, if you have.

AUT.- Both the university and government ethical committees have been referenced now.

REV.-  Please use the full name for ACL for the first time in line 76.

AUT.- It has been fixed.

REV.-  The cut points for different physical activity intensities you selected were made for children aged 5-8 years old (Ref. 31). Is it suitable for your participants aged 8-19 years old?

AUT.- The cut points utilized in this work were validated for ages between 5-15 [2](Trost, S. G.; Loprinzi, P. D.; Moore, R.; Pfeiffer, K. A., Comparison of accelerometer cut points for predicting activity intensity in youth. Med Sci Sports Exerc 2011,43, (7), 1360-1368). We acknowledge that we are using the criterion for older participants and we decided to use only one criterion to uniformize the methodology.In addition, previous studies with adolescents used Evenson´s cutoffs. A study with subjects from 13 to 15 years [3]and a second research from 6 to 17 years [4].

REV.-  How do you classify the normal weight and non-normal weight should be detailed in the method.

AUT.- The sample was split into a binary variable according to BMI age and specific-sex categories (normal weight and overweight/obese) and international cut-offs [5].

This method has been published elsewhere [6]

REV.-  In your study, should the VO2max be the dependent variable (line 108)?

AUT.- That is correct. It has been changed.

REV.-  Results

REV.-  My opinion is that if the moderate to vigorous physical activity (MVPA) has been reported, it is not necessary to report MPA and VPA separately again.

AUT.- Our rational to differentiate both components of MVPA was related with the fact that high intensity PA must have more consistent and robust associations with health indicators, including VO2max than low intensity PA in school-aged children and adolescents (Parikh, T.; Stratton, G. Sports Med, 2011, 41, (6), 477-488)(Poitras, V. J. et col. Appl Physiol Nutr Metab, 2016, 41, (6 Suppl 3), S197-239).

REV.-  Total MVPA time in Table 2 should be 3.4 min/recess rather than 2.7 min/recess, please double check all of your data. Moreover, the term should be consistent in the whole manuscript, for example, MVPA was used in Table 1 and 3, but MV in table 2.

AUT.- Thanks for catching this, we have reviewed the data.

REV.-  I think the title “The association between BMI, sedentary time and physical activity with VO2max” is clearer for Table 3. In addition, should the unit of most of the indicators in table 3 be min/recess rather than min/day.

AUT.- Thanks for the suggestions here, we are using your version now.

REV.-  I recommend you present the results of general linear model, which are more important than the results of correlation in Table 3. Furthermore, in order to obtain the independent association, other independent variables need to be adjusted. For example, daily ST and daily MVPA time without RT, and physical activity during RT (including LPA and MVPA) need to be adjusted, if you want to know the independent association between ST during RT and cardiorespiratory fitness (The publication PMID: 29221862 can be your reference).

AUT.- We are changing table 3 with correlations for table 3 general lineal model results. Regarding the statistics, we adjusted for total daily sedentary time and PA; however, adjusting for PA during recess time will bias the results because is the reciprocal of the RT and the effect of the PA during RT has been already removed when you adjust for TDPA without RT. Altogether, the extra adjustments would artificially  reduce the power of the analysis by introducing more variables than necessary in the model reducing the freedom degrees unnecessarily. Please, see section 2.3 Statistical analysis. We appreciate the bibliographic suggestion anyway.

REV.-  Considering age and sex are two significant predictors for VO2max, why performed the data analysis with different sex and ages together for table 3 and 4.

AUT.- Data in table 3 were adjusted for age, wear time and sex; these co-variates were also included in the full model. Nonetheless, we improved this analysis by included the general model table by age; in addition, results of the same analysis by school level were also summarized.

Regarding, the table 4 data are results of a logistic regression, which were already adjusted for age and sex. In addition, the cut values for VO2max are age and sex specific. In conclusion, our results are statistically independent of sex and age.

REV.-  For the table 4. a) The reference category for each independent variable need to be presented; b) the outcome indicator for the table 4 need to be noted (“High VO2max/ Healthy” or “Low VO2max / Unhealthy”); c) OR value cannot be negative, please double check your data; d) Health VO2max should be your outcome rather than the reference category.

AUT.- We have provided a more comprehensive table now.

REV.-  Discussion

The discussion section is weak. Furthermore, because of the main results probably are incorrect and need to be updated, the discussion may need to be rewritten.

AUT.- The results are not incorrect. All statistical procedures were carried out with conventional statistical packages and following specific literature in the field (for example, Kleinbaum DG, Kupper LL, Muller KE, Nizam A. Dummy Variables in Regression. Edtion ed. In: Kugushev A, ed. Applied Regression Analysis and Other Multivariable Methods: Duxbury Press, 1998:317-60). We recognize that all sections of the manuscript could be improved, including the discussion (this is inherent to all scientific literature); however, this does not imply our data or analysis are incorrect. Moreover, the work of this team is driven by high ethical standards both for field work and data analysis; therefore, our discussion and other sections tried to be informative and not speculative; indeed, as you can see in our Flux diagram (figure 1) and methods section, we used tight inclusion criteria (we even removed 26 participants due to extra analysis to avoid geographical bias), which is not the case in many publications nowadays. Our data included participants with at least 4 weekday, one weekend day, and 4 RT days, making our results strong, valid and reliable for the population included in the study.

Reviewer 2 Report

The study is well written, has an important clinical message, and should be of great interest to the readers of International Journal of Environment Research Public Health

The topic of this study is relevant, so it may be of interest to the readers. Some minor issues could be improved.

The manuscript can be further expanded and improved, and the reference list can be updated by citing recent studies about the topic.

Specially in introduction section. I suggest to include recent study reference in a different group for example in chronic patients, because authors have employed similar methods

I suggest include the following reference in introduction section

Brognara L, Navarro-Flores E, Iachemet L, Serra-Catalá N, Cauli O. Beneficial Effect of Foot Plantar Stimulation in Gait Parameters in Individuals with  Parkinson’s Disease. Brain Sci. 2020 Jan;10(2).

In addition, authors should review currently research about The Importance of Sedentary Time for and Improved VO2Max. Authors should also expand the limitations and future lines section suggesting different and novel approach to complement the findings of the present study.

As a physical activities and exercise expert, I really enjoyed reading it and feel it is novel work that adds to a growing area of research. As a reader it was enjoyable to read the article and as a reviewer, I could not suggest any more modifications in the content. I think that this manuscript is suitable for this journal and will be highly cited (when the minor revisions were worked).

Author Response

Dear editor, thank you very much for allowing to re-submit a revised version of the article. We have tried our best to accommodate and address the reviewers’ comments (please see text in changes mode).

REV.-  The study is well written, has an important clinical message, and should be of great interest to the readers of International Journal of Environment Research Public Health

The topic of this study is relevant, so it may be of interest to the readers. Some minor issues could be improved.

The manuscript can be further expanded and improved, and the reference list can be updated by citing recent studies about the topic.

Specially in introduction section. I suggest to include recent study reference in a different group for example in chronic patients, because authors have employed similar methods

I suggest include the following reference in introduction section

Brognara L, Navarro-Flores E, Iachemet L, Serra-Catalá N, Cauli O. Beneficial Effect of Foot Plantar Stimulation in Gait Parameters in Individuals with  Parkinson’s Disease. Brain Sci. 2020 Jan;10(2).

In addition, authors should review currently research about The Importance of Sedentary Time for and Improved VO2Max. Authors should also expand the limitations and future lines section suggesting different and novel approach to complement the findings of the present study.

As a physical activities and exercise expert, I really enjoyed reading it and feel it is novel work that adds to a growing area of research. As a reader it was enjoyable to read the article and as a reviewer, I could not suggest any more modifications in the content. I think that this manuscript is suitable for this journal and will be highly cited (when the minor revisions were worked).

 AUT.- Thanks for your time to review our manuscript and for your input. We have included your suggestions in the introduction and additional sentences in the limitations and future lines section.

Reviewer 3 Report

The main aim of the current manuscript was to explore associations between different variables of Physical activity (PA) and sedentary time (ST) during recess time at school and VO2max, of which they build up a relatively sound foundation in the introduction. The authors underline the lack of previous studies exploring this, especially among older students. However, their study sample consists of 172 students aged 8-19 years. This age difference among participants is problematic with regard to their analysis. The school days (both length and build-up of recess time, as well as outdoor area and available facilities for being physically active) are different in different age cohorts. In addition, the physical development children undergo from the age of 8-19 is substantial. Together, this contributes to the results being less informative than they would be with tighter age categories. Further, there are severe language/grammatical issues with the manuscript. Several points are made in long sentences and words are missing. In addition, there are many grammatical mistakes in the manuscript, especially with regard to the use of prepositions. The language issues in addition to the grammatical issues, make several parts of the manuscript hard to follow as a reader and may also introduce several misinterpretations. I believe that the aim of study is of relevance to explore further, as recess time provides possibilities of introducing PA that might contribute to the overall daily PA and in achieving daily PA recommendations. Nevertheless, I also acknowledge the large problems with the current version of the manuscript. I believe that the manuscript would be improved if it were reanalysed to be able to target different school levels (elementary-, lower secondary- and upper secondary school – or equivalent), and rewritten on the basis of the specific comments raised by reviewer.

  • Abstract:

    • Line 16: the authors should specify that they are referring to PA guidelines instead of using the phrase “health guidelines”.
    • Line 24-25: The authors state that “pupils who did not meet at least 15 min sedentary time during RT presented high probability of having unhealthy CRF”. This sentence can easily be misinterpreted as reporting that longer duration of sedentary time during RT is better.

    Introduction:

    • Line 42-44: This sentence makes little sense and I guess the authors meant to refer to “interventions to increase PA/MVPA” instead of “interventions to increase ST”.
    • Line 50-52: Makes a point about the gap in the knowledgebase when it comes to older students. However, the participants of the current study includes students form 8-19 years. (see comment regarding age group under “Methods”).

    Purpose:

    • OK.

    Methods:

    • Line 86-87: The authors state that the data collection protocol has been described elsewhere (ref. 29 and 30). Ref. 30 is a systematic review that addresses key information about data collection and -processing and do not report from the specific protocol of the current study. For a reader it is impossible to know which criteria is followed/discarded in the current study.
    • Year of data collection?
    • Participants: 172 pupils, aged 8-19 years. There is a very large age difference among participants. With regard to the physical development children undergo from the age of 8 to 19, in addition to the differences within the school days as a consequence of the different school levels they attend (amount and build-up of RT, outdoor area and available facilities for being physically active), I find it strange that these are all included in the same cohort. I suggest the authors do one of the following: 1. Reanalyse data in different age cohorts 2. Reanalyse data among the older students, removing the rest, as the authors state that the lack of knowledge exits particularly among older students (line 50-52). 3. Address the age issue and discuss potential consequences of including a study sample with such a large age spectrum.
      • However, because of the total amount of participants, I am unsure whether there are enough participants to perform alternative 1 and 2 without losing too much power in the analysis.
    • Attrition rate: The attrition rate was high (66,77 %). Where there performed any analysis to explore whether the ones with invalid measurements differed from the ones with valid measurements?
    • Accelerometer data processing: Insufficient amount of information about processing of the accelerometer data.
      • Which criteria was used to define non-wear?
      • The authors state that they’ve been using a 1 sec epoch. Were the Evenson’s cut points scaled up to match a 1 sec epoch?
      • Have intensity analyses been adjusted for wear time of the accelerometer?
    • Line 79-80: The abbreviation “CPX-T” is presented without any additional information or reference.
    • Line 97-99 information about test of cardiorespiratory fitness is provided. The author state that VO2max was measured among participants through an incremental step test. However, there is a discussion about ability to measure VO2max among young children, and often children’s peak oxygen consumption (VO2peak) is used instead (see for example: Resaland et al., 2009: doi: 10.1111/j.1600-0838.2009.01028.x) I recommend the choice of using VO2max instead of VO2peak to be addressed briefly in the methods section.
    • Covariates: The authors should consider to include a section about the covariates included in the study. Socio-economic status (SES) is not included as a covariate of the study. The influence of this variable on PA-level and cardiorespiratory fitness have been well- established. I therefore recommend this to be briefly addressed in the Strength and limitations section.
    • Statistical analyses: Participants were recruited from eight different schools. This might have a clustering effect in the results. PA might be more similar within schools/classes, for example. The authors provided no information about adjusting for this in their model. I suggest adding their arguments for doing so to the discussion section or reanalysing the data, adjusting for the clustering nature of the data.

    Results

    • Line 123: “[Boys] had a higher VO2max than girls”. As the onset of puberty influences VO2max trajectories for girls and boys substantially, the results from sex-specific analysis might differ between 8-year-old boys and girls compared to 19-year-old boys and girls. As this information is merely for descriptive purposes, the authors should consider addressing this briefly.
    • Table 1: These data are more or less meaningless presented as mean of the entire sample, as the age difference within the study sample is so large. This is also shown by very large SD. This table would be more informative if it was grouped after age categories in addition to Total, Boys and Girls.
      • “Total PA” is presented as “min/day” in Table 1. This measure of PA has not been described in the method section, and is merely described as a footnote in Table 1. This measure of PA must also be included in the method section.
      • Line 127-128: Information about statistical test performed. Delete, as it is sufficiently presented in 2.3 statistical analysis line 112-113.
    • Table 2: The same issue regarding study sample as in Table 1.
      • Moderate and Vigorous time are presented in minutes (min/recess) for different groups in Table 2. MV time is presented as the accumulation of minutes in Moderate to Vigorous (min/recess). However, when summing up the amount of time in categories Moderate and Vigorous this does not match the time presented in MV time. If these numbers are correct, an explanation for these differences ought to be included.
      • Line 139-140: Information about statistical test performed. Delete, as it is sufficiently presented in 2.3 statistical analysis line 112-113.
    • Table 3: OK, but there is a line under BMI that I believe should be removed.
    • Table 4: OK, but footnotes lack information in full of the abbreviations: MVPA Recess and NS.

    Discussion:

    • Section 187-193 discuss age as a significant predictor of VO2max, which is independent of total daily PA. The associations between age/maturation and VO2max is well established. Although appropriate to mention, these results do not add any new information. If the data were reanalysed within different age categories, the rate of change in VO2max due to age/maturation and due to change in ST (as briefly mentioned in line 191-192) could be addressed in more detail.
    • Line 195: The abbreviation “TDPA” is presented for the first time in the manuscript without any explanation as to what this abbreviation stands for.
    • Line 199: Provide information about author/s (not merely the reference number) when referring to their work in the sentence.
    • Line 218-219. Limitations of the accelerometer is superficially mentioned (…and so on.). Please be more specific about the limitations of the accelerometer and include relevant references for this.
    • Line 219. The authors state that the “study are only partially generalizable to the Spanish population”. This study collected data from a convenience sample from only eight schools in three different regions in Spain. In addition, the attrition rate was high. The generalizability of data to the Spanish population is therefore not possible to address

    Conclusion:

    • OK.

    References:

    • Reference 20 is insufficiently reported. Correct reference: Greca, J. P. de A., & Silva, D. A. S. (2017). Sedentary Behavior During School Recess in Southern Brazil. Perceptual and Motor Skills124(1), 105–117. https://doi.org/10.1177/0031512516681693.
    • Reference 27 needs to be altered as the world Medical Association (WMA) has been presented as “Association, W. M.”.

Author Response

Dear editor, thank you very much for allowing to re-submit a revised version of the article. We have tried our best to accommodate and address the reviewers’ comments (please see text in changes mode).

Authors appreciate your time and effort for such a detailed review of our manuscript with valuable comments and edits, which we believe can be accomplished to improve the understanding and quality of the manuscript.

REV.-  The main aim of the current manuscript was to explore associations between different variables of Physical activity (PA) and sedentary time (ST) during recess time at school and VO2max, of which they build up a relatively sound foundationin the introduction.

AUT.- Thanks for this comment.

REV.-  The authors underline the lack of previous studies exploring this, especially among older students. However, their study sample consists of 172 students aged 8-19 years. This age difference among participants is problematic with regard to their analysis. The school days (both length and build-up of recess time, as well as outdoor area and available facilities for being physically active) are different in different age cohorts. In addition, the physical development children undergo from the age of 8-19 is substantial. Together, this contributes to the results being less informative than they would be with tighter age categories.

AUT.- We understand your concern related with the range of age, and we partially share your opinion.

However, the main goal of this analysis was not to understand the reasons why children or adolescents engage in more or less activity during the recess, but to explore if amount of ST or PA period of daily time could be associated with an improved or impaired cardiorespiratory fitness. In addition to this, our analyses were always adjusted for age to rule out a plausible interaction between age, VO2max and PA/ST. Also, in the non-parametric statistics the VO2max dichotomization (healthy/unhealthy) is accounted for age as the cut-off values are sex and age specific. The recess time for the categories included in this study were the same for all schools and between middle and high school. For these previous reasons, we believe that the age factor is well considered in the manuscript, although we understand that additional analyses could have been done to be more specific about the influence of age/maturation in the RT behavior.

Nevertheless, your question motivated us to explore our environmental and weather logs during the weeks of assessment in each region. Therefore, these additional analysis by geographical regions made us to remove the students from the North region due to a significant difference in the weather between North and South regions (more rainy days in the North than in the South; this aspect could bias the amount of PA  and number of students who went outdoors during the recess time (the results did not change significantly). 

We have included additional sentences in the manuscript to clarify your concerns and improve the manuscript with your suggestions.

REV.-  Further, there are severe language/grammatical issues with the manuscript. Several points are made in long sentences and words are missing. In addition, there are many grammatical mistakes in the manuscript, especially with regard to the use of prepositions. The language issues in addition to the grammatical issues, make several parts of the manuscript hard to follow as a reader and may also introduce several misinterpretations.

AUT.- Sorry for all the language mistakes. We and an additional English native speaker have reviewed the second version again and we hope this version reads better.

REV.-  I believe that the aim of study is of relevance to explore further, as recess time provides possibilities of introducing PA that might contribute to the overall daily PA and in achieving daily PA recommendations. Nevertheless, I also acknowledge the large problems with the current version of the manuscript. I believe that the manuscript would be improved if it were reanalyzed to be able to target different school levels (elementary-, lower secondary- and upper secondary school – or equivalent) and rewritten on the basis of the specific comments raised by reviewer.

AUT.- Thanks again for your comments, we have made significant modifications in the manuscript following your and the other reviewers’ suggestions. However, we do not want to explore in deep differences between school levels, it was not our main goal here. Nonetheless, we need to say that after including school level in a GLM analysis, we found a couple of interesting results:

  • A significant interaction between recess ST x school level x sex. This analysis revealed that the relationship between sedentary time and VO2peak had significantly lower beta coefficient in girls than boys attending to high school, but not significant interaction was found in the other groups. We decided to add this information to results section.
  • Age was always a stronger predictor of VO2peak than age-category/school levelper se (adjusted R2 =0.287 for the model including groups of age (3 groups) compared with adjusted R2 =0.404 for the model including age as a continuous variable (the one presented in this manuscript).

REV.-  Abstract:

REV.-  Line 16: the authors should specify that they are referring to PA guidelines instead of using the phrase “health guidelines”.

AUT.- It has been corrected.

REV.-  Line 24-25: The authors state that “pupils who did not meet at least 15 min sedentary time during RT presented high probability of having unhealthy CRF”. This sentence can easily be misinterpreted as reporting that longer duration of sedentary time during RT is better.

AUT.- We rewrote appropriately.

Introduction:

REV.-  Line 42-44: This sentence makes little sense and I guess the authors meant to refer to “interventions to increase PA/MVPA” instead of “interventions to increase ST”.

AUT.- Thanks for noticing, we have corrected accordingly.

REV.-  Line 50-52: Makes a point about the gap in the knowledgebase when it comes to older students. However, the participants of the current study includes students form 8-19 years. (see comment regarding age group under “Methods”).

AUT.- We decided to include a wide range of age to address the gap across the whole spectrum of ages of middle and high school. Additional analysis have been included to clarify your questions (see other questions).

REV.-  Methods:

REV.-  Line 86-87: The authors state that the data collection protocol has been described elsewhere (ref. 29 and 30). Ref. 30 is a systematic review that addresses key information about data collection and -processing and do not report from the specific protocol of the current study. For a reader it is impossible to know which criteria is followed/discarded in the current study.

AUT.- We recognized this is confusing. Reference 30 has been deleted.

REV.-  Year of data collection?

AUT.- 2011-2013.

REV.-  Participants: 172 pupils, aged 8-19 years. There is a very large age difference among participants. With regard to the physical development children undergo from the age of 8 to 19, in addition to the differences within the school days as a consequence of the different school levels they attend (amount and build-up of RT, outdoor area and available facilities for being physically active), I find it strange that these are all included in the same cohort. I suggest the authors do one of the following: 1. Reanalyse data in different age cohorts 2. Reanalyse data among the older students, removing the rest, as the authors state that the lack of knowledge exits particularly among older students (line 50-52). 3. Address the age issue and discuss potential consequences of including a study sample with such a large age spectrum.

AUT.- Thanks for the suggestions. We offered a partial explanation related with the age issue in the comments section. Note that the main goal here is to explore if the PA and/or ST during RT can be associated with having a healthy or healthy VO2max. Although, we will not change the main goal of our paper; additional data by school level has been included (see previous comments). Altogether, we consider that importance of age has been well addressed in our data analysis: Firstly, age is included in all statistical models as a co-variate; in second, the healthy/unhealthy cut values are age-specific; in third, we did reanalyze this data for significant interaction between school-level (primary/middle/high-school groups)x sex x recess sedentary time to predict VO2max. The results GLM showed including age as continuous variable had the highest predictive power; as consequence, we decided to include age as a continuous variable and all school levels.

 All participants had similar facilities and the same RT across schools (playgrounds with basketball and soccer cohorts and large recreational areas), in Spain students between 8- and 18-years old share facilities, although the recess happens at different time of the day.

All data were collected in the same months of the year (September-October). In one of your previous questions we have recognize a plausible bias and it was fixed.

REV.-  However, because of the total amount of participants, I am unsure whether there are enough participants to perform alternative 1 and 2 without losing too much power in the analysis.

AUT.- Please, see previous comments.

REV.-  Attrition rate: The attrition rate was high (66,77 %). Where there performed any analysis to explore whether the ones with invalid measurements differed from the ones with valid measurements?

AUT.-Note that our attrition rate definition did not mean participants were not wearing their ACLs; instead, we used strict inclusion criteria to select the final dataset. Indeed, if we had followed the epidemiological criteria reported in the literature for traditional PA analysis our final sample size would have tripled (1 weekday and 1 weekend day).

Initially, we recruited a total of 668 students for a more comprehensive study. After considering: Lost data (33 accelerometers), lost device (3 accelerometers) and meeting Troiano, et al. [7]282 accelerometers reported valid data for usual PA patterns/levels analysis. However, the extra criterion of 4 recess days reduced the sampleto172 subjects; which was also more limited after considering geographical bias (see comment above). 

Maybe the attrition word was not appropriate to describe our final inclusion rate.

REV.-  Accelerometer data processing: Insufficient amount of information about processing of the accelerometer data.

AUT.- We have added information in this regard.

REV.-  Which criteria was used to define non-wear?

AUT.- The criteria wasTroiano, Berrigan and Dodd [7]criteria of ≥10 valid wear hours per day.

Briefly, a period of 60 minutes of consecutive zeros, allowing for 2 minutes of non-zero interruptions, anywhere in the data array was considered as non-wear time.

REV.-  The authors state that they’ve been using a 1 sec epoch. Were the Evenson’s cut points scaled up to match a 1 sec epoch?

AUT.- Evenson [8]used a 15 sec epoch. We recognize that longer epochs must reduce accuracy; however, shorter periods must not presume a limitation.

REV.-  Have intensity analyses been adjusted for wear time of the accelerometer?

AUT.- Yes, they did. Sorry, we have cut a lot of information because of word limit requirements. More comprehensive description of the methodology and statistical analysis has been included now.

REV.-  Line 79-80: The abbreviation “CPX-T” is presented without any additional information or reference.

AUT.- We apologize for this mistake, we cut a lot of information because of word limit requirements. We have included proper description.

REV.-  Line 97-99 information about test of cardiorespiratory fitness is provided. The author state that VO2max was measured among participants through an incremental step test. However, there is a discussion about ability to measure VO2max among young children, and often children’s peak oxygen consumption (VO2peak) is used instead (see for example: Resaland et al., 2009: doi: 10.1111/j.1600-0838.2009.01028.x) I recommend the choice of using VO2max instead of VO2peak to be addressed briefly in the methods section.

AUT.- We understand your point in the terminology, and we do not have any problem in using VO2peak instead VO2max.

 REV.-  Covariates: The authors should consider to include a section about the covariates included in the study. Socio-economic status (SES) is not included as a covariate of the study. The influence of this variable on PA-level and cardiorespiratory fitness have been well- established. I therefore recommend this to be briefly addressed in the Strength and limitations section.

AUT.- We have included it in the limitations section. However, we want to reinforce that it was not our goal to focus on factors that can modify the independent variables of this study, but just confirm if a certain amount of PA/ST accumulated during the RT were associated with a healthy/unhealthy VO2peak independently of TDPA, ST and the main biological factors (age and sex). Nonetheless, we share with you the fact SES are highly relevant variables to include in future behavioral studies.

REV.-  Statistical analyses: Participants were recruited from eight different schools. This might have a clustering effect in the results. PA might be more similar within schools/classes, for example. The authors provided no information about adjusting for this in their model. I suggest adding their arguments for doing so to the discussion section or reanalysing the data, adjusting for the clustering nature of the data.

AUT.- By law, all government schools in Spain must have similar schedules and facilities. Moreover, we confirmed all schools in our study had 2 weekly physical education classes, and one recess of 30 minutes per day. Nonetheless, we provide descriptive statistics about the PA, ST, morphology and VO2peak across schools for informative purposes.

REV.-  Results

Line 123: “[Boys] had a higher VO2max than girls”. As the onset of puberty influences VO2max trajectories for girls and boys substantially, the results from sex-specific analysis might differ between 8-year-old boys and girls compared to 19-year-old boys and girls. As this information is merely for descriptive purposes, the authors should consider addressing this briefly.

AUT.- Thanks for the suggestion. We addressed this aspect in the new version.

REV.-  Table 1: These data are more or less meaningless presented as mean of the entire sample, as the age difference within the study sample is so large. This is also shown by very large SD. This table would be more informative if it was grouped after age categories in addition to Total, Boysand Girls.

AUT.- We have included a table by age categories.

REV.-  “Total PA” is presented as “min/day” in Table 1. This measure of PA has not been described in the method section, and is merely described as a footnote in Table 1. This measure of PA must also be included in the method section.

AUT.- We did add this category and provided appropriate description.

REV.-  Line 127-128: Information about statistical test performed. Delete, as it is sufficiently presented in 2.3 statistical analysis line 112-113.

AUT.- We think this information is relevant since Mann-Whitney-U test is a non-parametric test and table shows mean and standard deviations.

REV.-  Table 2: The same issue regarding study sample as in Table 1.

AUT.- The same answer as above.

REV.-  Moderate and Vigorous time are presented in minutes (min/recess) for different groups in Table 2. MV time is presented as the accumulation of minutes in Moderate to Vigorous (min/recess). However, when summing up the amount of time in categories Moderate and Vigorous this does not match the time presented in MV time. If these numbers are correct, an explanation for these differences ought to be included.

AUT.- Thanks for catching this, we have reviewed the data.

REV.-  Line 139-140: Information about statistical test performed. Delete, as it is sufficiently presented in 2.3 statistical analysis line 112-113.

AUT.- See previous answers please.

REV.-  Table 3: OK, but there is a line under BMI that I believe should be removed.

AUT.- Thanks. Line has been removed.

REV.-  Table 4: OK, but footnotes lack information in full of the abbreviations: MVPA Recess and NS.

AUT.- Thanks for noticing, it has been included.

REV.-   Discussion:

REV.-  Section 187-193 discuss age as a significant predictor of VO2max, which is independent of total daily PA. The associations between age/maturation and VO2max is well established. Although appropriate to mention, these results do not add any new information. If the data were reanalysed within different age categories, the rate of change in VO2max due to age/maturation and due to change in ST (as briefly mentioned in line 191-192) could be addressed in more detail.

AUT.- We understand your concern. The potential influence of age on the relationship between VO2max and PA or ST, was addressed by adjusting all analyses for age. In a previous analysis of this dataset, we did find an interaction between school periods, sex (primary/middle/high-school), ST (see previous comments).

REV.-  Line 195: The abbreviation “TDPA” is presented for the first time in the manuscript without any explanation as to what this abbreviation stands for.

AUT.- Thanks, this abbreviation has been included in the methods section.

REV.-  Line 199: Provide information about author/s (not merely the reference number) when referring to their work in the sentence.

AUT.- We have referenced the first author now.

REV.-  Line 218-219. Limitations of the accelerometer is superficially mentioned (…and so on.). Please be more specific about the limitations of the accelerometer and include relevant references for this.

AUT.- Accelerometers limitations have been included along new references.

REV.-  Line 219. The authors state that the “study are only partially generalizable to the Spanish population”. This study collected data from a convenience sample from only eight schools in three different regions in Spain. In addition, the attrition rate was high. The generalizability of data to the Spanish population is therefore not possible to address

AUT.- Thanks for the recommendation. Nonetheless, we did mean to say our study was in fact partially generalizable, because it included only 8 schools and three regions (now only six). We have rewritten the sentence for a better understanding.

REV.-  References:

Reference 20 is insufficiently reported. Correct reference: Greca, J. P. de A., & Silva, D. A. S. (2017). Sedentary Behavior During School Recess in Southern Brazil. Perceptual and Motor Skills124(1), 105–117. https://doi.org/10.1177/0031512516681693.

Reference 27 needs to be altered as the World Medical Association (WMA) has been presented as “Association, W. M.”.

AUT.- Thanks for noticing, we have corrected appropriately.

NEW REFERENCES

  1. Van Kann, D. H. H.; de Vries, S. I.; Schipperijn, J.; de Vries, N. K.; Jansen, M. W. J.; Kremers, S. P. J., A Multicomponent Schoolyard Intervention Targeting Children's Recess Physical Activity and Sedentary Behavior: Effects After 1 Year. Journal of physical activity & health 2017,14, (11), 866-875.
  2. Trost, S. G.; Loprinzi, P. D.; Moore, R.; Pfeiffer, K. A., Comparison of accelerometer cut points for predicting activity intensity in youth. Med Sci Sports Exerc 2011,43, (7), 1360-1368.
  3. Aibar, A.; Bois, J. E.; Generelo, E.; Zaragoza Casterad, J.; Paillard, T., A cross-cultural study of adolescents' physical activity levels in France and Spain. Eur J Sport Sci 2013a,13, (5), 551-558.
  4. Adams, M. A.; Johnson, W. D.; Tudor-Locke, C., Steps/day translation of the moderate-to-vigorous physical activity guideline for children and adolescents. Int J Behav Nutr Phys Act 2013,10, (1), 49.
  5. Cole, T. J.; Bellizzi, M. C.; Flegal, K. M.; Dietz, W. H., Establishing a standard definition for child overweight and obesity worldwide: international survey. Bmj 2000,320, (7244), 1240.
  6. Calahorro-Canada, F.; Torres-Luque, G.; Lopez-Fernandez, I.; Carnero, E. A., Is physical education an effective way to increase physical activity in children with lower cardiorespiratory fitness? Scandinavian journal of medicine & science in sports 2017,27, (11), 1417-1422.
  7. Troiano, R. P.; Berrigan, D.; Dodd, K., Physical Activity in the United States Measured by Accelerometer. Med Sci Sports Exerc 2008,40, (1), 181-188.
  8. Evenson, K. R.; Catellier, D. J.; Gill, K.; Ondrak, K. S.; McMurray, R. G., Calibration of two objective measures of physical activity for children. J Sports Sci 2008,24, (14), 1557-1565.

Round 2

Reviewer 1 Report

The authors have addressed some comments made in my previous review. Whereas, some comments still need to be considered.

Major comments

  1. In the first round of review, I proposed that the OR (odds ratio) value should not be less than 0, so the main result of this manuscript OR = -3.7 is incorrect. The author's reply is that logitOR could be less than 1, I guess authors want to say that logitOR could be less than 0. Indeed, logitOR can be less than 0 is correct, and logitOR = beta coefficient. But, OR = exp (beta) = exp (logitOR), no doubt, it is impossible to be less than 0. Although both beta and OR are two results of logistic regression, most of the publications only report OR values as the final result, because beta is inconvenient for the interpretation of results. Although the description of the range of logitOR and OR in author’s reply is correct, please note that your initial manuscript reports the OR value, not the logitOR. And, LogitOR and OR are totally different. Furthermore, authors also can know the direction of the relationship between dependent and independent variables, by comparing the OR value with 1. (Authors can read that how to calculate and understand the LogitOR and OR in the link below, https://stats.idre.ucla.edu/other/mult-pkg/faq/general/faq-how-do-i-interpret-odds-ratios-in-logistic-regression/).
  2. In this version, the main results also have some problems. For example, in Table 4, the OR value for ST Recess> 15min equals 0.02, the author’s explanation is "children and adolescents who accumulate more than 15 minutes of ST during recess had a low likelihood (2.2%) to have a healthy cardiorespiratory fitness". Two details are needed to be highlighted here: 1) The data is inconsistent. OR = 0.02 in Table 4, OR = 0.021 in the abstract, and 2.2% for the description in Line 209; 2) Data interpretation is probably inaccuracy. I recommend that the exact interpretation for OR=0.02 should be “Compared to individuals accumulate less than 15-min of ST during recess, individuals accumulate more than 15-min of ST during recess had a 98% (1-0.02) lower odds to achieve healthy cardiorespiratory fitness” (Publication in Circulation [PMID: 29511001] can be your reference). In addition, please double check whether the value of OR=0.02 is correct, because this value is really small.
  3. Authors found the relationship between sedentary time and VO2peak had significant interaction between recess ST x School level x Sex, but the analyses were not conducted separately by school level and sex group, the insufficient sample size may be one of the important reasons. Based on the interaction effect, I don't think it is reasonable to combine the data to do the data analysis, although authors considered these variables are confounders. Obviously, the insufficient sample size and the large range age span are one of the major limitations of this study. Furthermore, the failure to analyze by age and sex may be one of the important reasons why there is no significant association between MVPA in the Recess and CRF.

Minor comments

  1. Spelling "adolecents" wrong in the abstract, should be “adolescents”.
  2. The conclusion in the abstract, authors suggested that “school-aged children and adolescents must be empowered to perform PA during RT to prevent deleterious effects of sedentary time on CRF”. But, the results of this study indicated there is no significant association between MVPA in the Recess and CRF. How do the authors view this contradictory issue?
  3. Figure 1 and Table 1 should be displayed on the same page;
  4. Tables 3 and 4 can be combined. The authors conducted category and linear analyses to increase the robustness of the main results, which is very good, I suggest that combine them into one table to display concisely (Table 2 the in publication PMID: 28693036 can be your reference);
  5. Although in the method section introduces the variables used as adjustments in the statistics, they are not listed in Tables 3 and 4, I suggest to add this information. Because many readers may directly read your main results (figures and tables) instead of reading your full text, which will make readers miss some key information.
  6. Considered the p value for ST_woRecess was significant in Table 4, the OR value should not be equal to 1, as well as 95% CI for ST_woRecess should not include 1. As mentioned in major comments, the value of OR >1 or < 1 represent different direction for the association between dependent and independent variables. Based on the logitOR=-0.004, OR should be exp(logit OR)=exp(-0.004)=0.996. Also, authors can report 95% CI to 3 decimal places to present the exact value.
  7. I guess in Line 202 should have two beta coefficients, one for boys and one for girls; or listed the results of p for interaction.
  8. The sample size should be the main limitation of this study, but it was not mentioned in the limitation section.

Author Response

Dear editor, thank you very much for allowing to re-submit a revised version of the article. We have tried our best to accommodate and address the reviewers’ comments (please see text in changes mode).

REV.- In the first round of review, I proposed that the OR (odds ratio) value should not be less than 0, so the main result of this manuscript OR = -3.7 is incorrect. The author's reply is that logitOR could be less than 1, I guess authors want to say that logitOR could be less than 0. Indeed, logitOR can be less than 0 is correct, and logitOR = beta coefficient. But, OR = exp (beta) = exp (logitOR), no doubt, it is impossible to be less than 0. Although both beta and OR are two results of logistic regression, most of the publications only report OR values as the final result, because beta is inconvenient for the interpretation of results. Although the description of the range of logitOR and OR in author’s reply is correct, please note that your initial manuscript reports the OR value, not the logitOR. And, LogitOR and OR are totally different. Furthermore, authors also can know the direction of the relationship between dependent and independent variables, by comparing the OR value with 1. (Authors can read that how to calculate and understand the LogitOR and OR in the link below, https://stats.idre.ucla.edu/other/mult-pkg/faq/general/faq-how-do-i-interpret-odds-ratios-in-logistic-regression/).

AUT.- Thanks again for your comments and we share all of them; we recognized our mistake of exchanging logitOR by OR in our answers to you. Note we always mention that our values came from logistic regression.

REV.- In this version, the main results also have some problems. For example, in Table 4, the OR value for ST Recess> 15min equals 0.02, the author’s explanation is "children and adolescents who accumulate more than 15 minutes of ST during recess had a low likelihood (2.2%) to have a healthy cardiorespiratory fitness". Two details are needed to be highlighted here: 1) The data is inconsistent. OR = 0.02 in Table 4, OR = 0.021 in the abstract, and 2.2% for the description in Line 209; 2) Data interpretation is probably inaccuracy. I recommend that the exact interpretation for OR=0.02 should be “Compared to individuals accumulate less than 15-min of ST during recess, individuals accumulate more than 15-min of ST during recess had a 98% (1-0.02) lower odds to achieve healthy cardiorespiratory fitness” (Publication in Circulation [PMID: 29511001] can be your reference). In addition, please double check whether the value of OR=0.02 is correct, because this value is really small.

AUT.- Data are not inconsistent; their format can be inconsistent. In the abstract there are 3 decimal places and 2 decimal places in the table 4. We have put 3 decimal places in table 4 now.

Regarding to interpretation, there are only two possible ways of reading OR, the probability that the event occurs or not occur, which is always reciprocal between groups. 

Indeed, the value is small, and it has been checked several times.

REV.- Authors found the relationship between sedentary time and VO2peak had significant interaction between recess ST x School level x Sex, but the analyses were not conducted separately by school level and sex group, the insufficient sample size may be one of the important reasons. Based on the interaction effect, I don't think it is reasonable to combine the data to do the data analysis, although authors considered these variables are confounders. Obviously, the insufficient sample size and the large range age span are one of the major limitations of this study. Furthermore, the failure to analyze by age and sex may be one of the important reasons why there is no significant association between MVPA in the Recess and CRF.

AUT.- Again, in our point of view your suggestion would tend to overparameterize the model:

  • The dependent variable (VO2 category) was created considering cut values specific for sex and age, which makes their effect is partially included in the classification H or UH.
  • We do not need to split the analysis per school level because our main hypothesis is that the ST during RT will be related with the provability to have healthy or unhealthy VO2peak. If this categorization is not enough, it would be the same that recognize that the international cut values are erroneous, which would be a question out of the scope of this study.
  • Moreover, the general lineal model which took in consideration school-level had lower coefficient of determination than the one including age as co-variates. By including sex, both in the GLM and logistic regression we are already analyzing the effect of the variables.
  • The large range of age span is positive aspect of the study. It demonstrates that, the association between VO2peak categories and Recess ST is modulated by age and independent of it.
  • We recognize than a larger sample size could increase the power of some or our analyses. Also, it would be necessary a larger sample size to prove some of the hypothesis you suggest.
  • We understand that data by age categories could be of interest for descriptive purposes and we have included this data in table 1 and results from the additional analysis suggested by you in the first review. We think that this will be enough to cover this topic.

REV.- Spelling "adolecents" wrong in the abstract, should be “adolescents”.

AUT.- It has been corrected

REV.- The conclusion in the abstract, authors suggested that “school-aged children and adolescents must be empowered to perform PA during RT to prevent deleterious effects of sedentary time on CRF”. But, the results of this study indicated there is no significant association between MVPA in the Recess and CRF. How do the authors view this contradictory issue?

AUT.- We recommend PA in general, not necessarily MVPA. PA is any type of muscle contraction above the resting values, including light physical activity; this is what we are recommending.

AUT.- In fact, the relevance of the light physical activity has been highlighted many times in the literature.   

REV.- Figure 1 and Table 1 should be displayed on the same page;

Tables 3 and 4 can be combined. The authors conducted category and linear analyses to increase the robustness of the main results, which is very good, I suggest that combine them into one table to display concisely (Table 2 the in publication PMID: 28693036 can be your reference);

AUT.- We have combined figure 1 and table 1 together.

Regarding table 3 and 4, we don’t think that mixing parametric and non-parametric results in the same table help with the understanding of the results. In fact, we have been criticized in the past for other reviewers. Anyway, we appreciate your suggestion and comments.

REV.- Although in the method section introduces the variables used as adjustments in the statistics, they are not listed in Tables 3 and 4, I suggest to add this information. Because many readers may directly read your main results (figures and tables) instead of reading your full text, which will make readers miss some key information.

AUT.- Two paragraphs have been included, each one before each table.

REV.- Considered the p value for ST_woRecess was significant in Table 4, the OR value should not be equal to 1, as well as 95% CI for ST_woRecess should not include 1. As mentioned in major comments, the value of OR >1 or < 1 represent different direction for the association between dependent and independent variables. Based on the logitOR=-0.004, OR should be exp(logit OR)=exp(-0.004)=0.996. Also, authors can report 95% CI to 3 decimal places to present the exact value.

AUT.- Three decimal places are reported now.

REV.- I guess in Line 202 should have two beta coefficients, one for boys and one for girls; or listed the results of p for interaction.

AUT.- The boys’ coefficient is the reference. As you may well-known for binary variables there is one beta coefficient. The beta coefficient indicates that girls’ VO2max was 6.67 ml/kg/min lower than boys on average (in this sample).

REV.- The sample size should be the main limitation of this study, but it was not mentioned in the limitation section.

AUT.- We have included this in the limitations. However, it is not a limitation for our logistic regression analysis, which is the main focus of our statistical analysis.

Reviewer 3 Report

Brief summary:

General impression is that the alterations the authors have conducted since their first submission have elevated the quality of the manuscript substantially. The authors have also provided sound explanations for most of the reviewer comments they chose to discard. However, there are still a lot of very long sentences which are hard to follow, in addition to several grammatical errors in the manuscript. I would recommend you to get some additional help with this.

Abstract:

  • OK

Introduction:

  • Look specifically at the grammar in line 37-40. Divide into two sentences instead? “MVPA (4,6). Breaking ST (…)”.
  • Look specifically at the grammar in line 48. “..it has reported that can contribute up to 40 %..”. Use "and has been reported to contribute up to 40 %..." instead?
  • Look specifically at the grammar in line 55. “…cutoffs healthy…”. Use “cutoffs for health..” instead?
  • Look specifically at the grammar in line 61-63. In my opinion, some words are lacking: “…is a question has not been answered…”

Purpose:

  • OK.

Methods:

  • Information about year of data collection should be included in one of the first sentences under 2.1.
  • My comment regarding age difference among participants is thoroughly addressed by the authors. I understand their arguments and believe that the clarifications they have made through response to reviewer, some analyses, data presentation and text is sufficient.
  • 2 Procedures: I find your additional information regarding selected/non-selected participants highly relevant and satisfactory in order to gain a better understanding of your attrition rates.
  • Line 87: Grammatical error; North is spelled incorrectly.
  • Line 90: Throughout the entire introduction the authors use the term ‘VO2max’. However, in line 90 they refer to ‘VO2peak’ without any additional information regarding the two measurements. Either include a sentence to explain what VO2peak is compared with VO2max, or VO2peak should be used throughout the article for better consistency (of course with the exception of referring to data from other articles that have measured VO2max).
  • In line 102 you state that you have used a 1 sec epoch. You further state that you have used Evenson’s cut points of 15 sec. In my experience, ActiLife will, by default, use a 15 sec epoch in the intensity analyses to match the cut points by Evenson. This means that all intensity analysis will be based on a 15 sec epoch unless you have manually created an equation were you have scaled up the cut points to match a 1 sec epoch. (See for example: Dalene, et al. 2018; doi:10.1186/s12889-018-5610-7 for more information about this). My question in the first revision related to this point and was not a critique of the epoch used.
  • In line 103-104 you write that “A wear-time (…) for ≥5 days (…)”. In line 117-118 you write that “A wear-time (…) for ≥4 days (…). As duplicates, one of these sentences must be deleted (I suggest the latter). In addition, the correct wear-criteria must be presented.
  • I find the two sections starting in line 98 and in line 111 a bit unstructured. I suggest that you move the information about accelerometers (placement and instructions for use) somewhere after “briefly” in line 100, were the rest of the accelerometer information are presented.
  • The information regarding the protocol for measuring VO2peak is more comprehensive and sufficiently addressed.
  • Line 159: For improved readability, I suggest dividing the long sentence into two sentences between “categories” and “for”.

Results

  • In the first revision, I commented on the relevance of data in table 1 (where the descriptive characteristics of the study sample merely are presented for the entire age group). In your reply to this comment, you say that you have included a table by age categories. However, I am unable to find this in the revised version. I would strongly advise you to include age categories in the descriptive table in order to make the descriptive statistics informative.
  • Line 179-180: you may want to restructure the sentence a bit, as the meaning of the last part is somewhat unclear.
  • Line 187: Grammatical error; between is spelled
  • Line 193 (heading for table 3): grammatical error; estimate is spelled incorrectly.
  • Look specifically at the grammar in line 202-203. Divide into two sentences instead? “(…) 0.001. No significant interaction (…)”.
  • Line 208-209: Grammatical error; recess is spelled incorrectly. In addition, include “that” between “confirmed” “children and adolescents”?
  • Line 216-217; remove “did not enter” as it is already said earlier in that sentence.

Discussion:

  • Look specifically at the grammar in line 220-223. Divide into two sentences instead? “(…) adolescents. We also found (…)”.
  • Look specifically at the grammar in line 224-227. Divide into two sentences instead?
  • Line 228: Include “whether” between “confirm” and “ specific”.
  • Line 239: Grammatical error; choose either “should positively associate with (…)” or “should be positively associated with (…)”.
  • Line 258-259: Grammatical error; both “typically” and unstructured” are spelled incorrectly.
  • Line 266-269: Long sentence. Divide into two sentences instead?
  • Line 273-278: Long sentence. Divide into two or more sentences instead?
  • Line 278-282: Long sentence. Consider revising the sentence somewhat, and divide into two or more sentences instead?
  • Line 278-279: As commented in the first review, I do not think “partially generalizable” is an appropriate word to use when the study population is product of a convenience sample, and only included 6 schools. I would suggest to say something like: “As the study sample consist of a convenience sample of six schools in two regions of Spain, generalizability of results to the Spanish population cannot be inferred”.
  • Line 285-290: Long sentence. Consider revising the sentence somewhat, and divide into two or more sentences instead.

Conclusion:

OK.

References:

OK.

Author Response

Dear editor, thank you very much for allowing to re-submit a revised version of the article. We have tried our best to accommodate and address the reviewers’ comments (please see text in changes mode).

Brief summary:

General impression is that the alterations the authors have conducted since their first submission have elevated the quality of the manuscript substantially. The authors have also provided sound explanations for most of the reviewer comments they chose to discard. However, there are still a lot of very long sentences which are hard to follow, in addition to several grammatical errors in the manuscript. I would recommend you to get some additional help with this.

Introduction:

REV.- Look specifically at the grammar in line 37-40. Divide into two sentences instead? “MVPA (4,6). Breaking ST (…)”.

AUT.- We divided into 2 sentences

REV.- Look specifically at the grammar in line 48. “..it has reported that can contribute up to 40 %..”. Use "and has been reported to contribute up to 40 %..." instead?

AUT.- Thanks, it has been modified.

REV.- Look specifically at the grammar in line 55. “…cutoffs healthy…”. Use “cutoffs for health..” instead?

AUT.- We are refering to a “healthy CRF” profile. In order to provide a better understanding we included “profile” after CRF.

REV.- Look specifically at the grammar in line 61-63. In my opinion, some words are lacking: “…is a question has not been answered…”

Methods:

REV.- Information about year of data collection should be included in one of the first sentences under 2.1.

AUT.- We included this information

REV.- My comment regarding age difference among participants is thoroughly addressed by the authors. I understand their arguments and believe that the clarifications they have made through response to reviewer, some analyses, data presentation and text is sufficient.

AUT.- Thank you

REV.- 2 Procedures: I find your additional information regarding selected/non-selected participants highly relevant and satisfactory in order to gain a better understanding of your attrition rates.

AUT.- Thank you

REV.- Line 87: Grammatical error; North is spelled incorrectly.

AUT.- Corrected

REV.- Line 90: Throughout the entire introduction the authors use the term ‘VO2max’. However, in line 90 they refer to ‘VO2peak’ without any additional information regarding the two measurements. Either include a sentence to explain what VO2peak is compared with VO2max, or VO2peak should be used throughout the article for better consistency (of course with the exception of referring to data from other articles that have measured VO2max).

AUT.- According to the suggestion, we have used VO2peak throughout the article for better consistency, with the exception of referring to data from two of the articles that have measured VO2max.

REV.- In line 102 you state that you have used a 1 sec epoch. You further state that you have used Evenson’s cut points of 15 sec. In my experience, ActiLife will, by default, use a 15 sec epoch in the intensity analyses to match the cut points by Evenson. This means that all intensity analysis will be based on a 15 sec epoch unless you have manually created an equation were you have scaled up the cut points to match a 1 sec epoch. (See for example: Dalene, et al. 2018; doi:10.1186/s12889-018-5610-7 for more information about this). My question in the first revision related to this point and was not a critique of the epoch used.

AUT.- Excuse for our misunderstanding. We confirm that ActiLife reintegrates 1 sec epoch to 15 sec epoch to match the Evenson’s cut points.

REV.- In line 103-104 you write that “A wear-time (…) for ≥5 days (…)”. In line 117-118 you write that “A wear-time (…) for ≥4 days (…). As duplicates, one of these sentences must be deleted (I suggest the latter). In addition, the correct wear-criteria must be presented.

AUT.- It has been modified.

REV.- I find the two sections starting in line 98 and in line 111 a bit unstructured. I suggest that you move the information about accelerometers (placement and instructions for use) somewhere after “briefly” in line 100, were the rest of the accelerometer information are presented.

AUT.- It has been modified.

REV.- The information regarding the protocol for measuring VO2peak is more comprehensive and sufficiently addressed.

AUT.- thanks

REV.- Line 159: For improved readability, I suggest dividing the long sentence into two sentences between “categories” and “for”.

AUT.- It has been divided.

Results

REV.- In the first revision, I commented on the relevance of data in table 1 (where the descriptive characteristics of the study sample merely are presented for the entire age group). In your reply to this comment, you say that you have included a table by age categories. However, I am unable to find this in the revised version. I would strongly advise you to include age categories in the descriptive table in order to make the descriptive statistics informative.

AUT.- A new table has been included. And also a new paragraph in the method section (lines 158-160).

REV.- Line 179-180: you may want to restructure the sentence a bit, as the meaning of the last part is somewhat unclear.

AUT.- It has been re-written.

REV.- Line 187: Grammatical error; between is spelled

AUT.- Thanks. It has been corrected.

REV.- Line 193 (heading for table 3): grammatical error; estimate is spelled incorrectly.

AUT.- Thanks. It has been corrected.

REV.- Look specifically at the grammar in line 202-203. Divide into two sentences instead? “(…) 0.001. No significant interaction (…)”.

AUT.- It has been divided.

REV.- Line 208-209: Grammatical error; recess is spelled incorrectly. In addition, include “that” between “confirmed” “children and adolescents”?

AUT.- Included.

REV.- Line 216-217; remove “did not enter” as it is already said earlier in that sentence.

AUT.- Thanks, removed.

Discussion:

REV.- Look specifically at the grammar in line 220-223. Divide into two sentences instead? “(…) adolescents. We also found (…)”.

AUT.- It has been divided.

REV.- Look specifically at the grammar in line 224-227. Divide into two sentences instead?

AUT.- It has been divided.

REV.- Line 228: Include “whether” between “confirm” and “ specific”.

AUT.- Thanks. Included.

REV.- Line 239: Grammatical error; choose either “should positively associate with (…)” or “should be positively associated with (…)”.

AUT.- Thanks. Corrected

REV.- Line 258-259: Grammatical error; both “typically” and unstructured” are spelled incorrectly.

AUT.- Thanks. Corrected

REV.- Line 266-269: Long sentence. Divide into two sentences instead?

AUT.- It has been divided.

REV.- Line 273-278: Long sentence. Divide into two or more sentences instead?

AUT.- It has been divided and re-written.

REV.- Line 278-282: Long sentence. Consider revising the sentence somewhat, and divide into two or more sentences instead?

AUT.- It has been divided and re-written.

REV.- Line 278-279: As commented in the first review, I do not think “partially generalizable” is an appropriate word to use when the study population is product of a convenience sample, and only included 6 schools. I would suggest to say something like: “As the study sample consist of a convenience sample of six schools in two regions of Spain, generalizability of results to the Spanish population cannot be inferred”.

AUT.- Modified.

REV.- Line 285-290: Long sentence. Consider revising the sentence somewhat, and divide into two or more sentences instead.

AUT.- This part has been re-written.

Round 3

Reviewer 1 Report

The authors have addressed the comments made in my previous review carefully. I have just one major comment and one minor comment:

Major comments

Again, the author’s interpretation of the main result of OR = 0.023 is still inaccurate.

It should be noted that probability and OR were quite different. First, the concept is different. Probability (p) is an absolute term, which always ranges between 0 and 1. Inversely, OR is a relative term, which ranges from >0 to infinity. In addition, odds = p / (1-p). OR = odds (group A) / odds (group B, as a reference). Second, as a relative variable, a reference group is needed when interpreting OR.

Therefore, I suggest the exact interpretation for OR=0.023 should be “Compared to individuals accumulate less than 15-min of ST during recess, individuals accumulate more than 15-min of ST during recess had a 97.7% lower odds to achieve healthy cardiorespiratory fitness”. (Publication in Circulation [PMID: 29511001] can be your reference). The description in abstract and line 217-218 need to be revised.

Furthermore, I recommend the outcome (or event) to be changed from health CRF to unhealthy CRF in Table 4, which makes the results easier to understand. If so, your main result of OR for “ST Recess>15min” is probably 43.47 (1/0.023), which means engaging in more sedentary time in recess (> 15-min vs. ≤15-min) are more likely to get unhealthy CRF.

Minor comments

  1. The results of logitOR are not necessary to be presented in the manuscript.

Author Response

Dear editor, thank you very much for allowing to re-submit a revised version of the article. We have tried our best to accommodate and address the reviewers’ comments (please see text in changes mode).

AUT. Thanks for your final suggestions. We have made appropriate corrections to follow your recommendations.
